# Kauniolide synthase is a P450 with unusual hydroxylation and cyclization-elimination activity

Qing Liu[1,2], Arman Beyraghdar Kashkooli[1], David Manzano[3,4], Irini Pateraki [2], Lea Richard[1], Pim Kolkman[1], Maria Fátima Lucas[5], Victor Guallar[5,6], Ric C.H. de Vos[7], Maurice C.R. Franssen[8], Alexander van der Krol [1] & Harro Bouwmeester [1,9]

Guaianolides are an important class of sesquiterpene lactones with unique biological and pharmaceutical properties. They have been postulated to be derived from germacranolides, but for years no progress has been made in the elucidation of their biosynthesis that requires an unknown cyclization mechanism. Here we demonstrate the isolation and characterization of a cytochrome P450 from feverfew (*Tanacetum parthenium*), kauniolide synthase. Kauniolide synthase catalyses the formation of the guaianolide kauniolide from the germacranolide substrate costunolide. Unlike most cytochrome P450s, kauniolide synthase combines stereoselective hydroxylation of costunolide at the C3 position, with water elimination, cyclization and regioselective deprotonation. This unique mechanism of action is supported by in silico modelling and docking experiments. The full kauniolide biosynthesis pathway is reconstructed in the heterologous hosts *Nicotiana benthamiana* and yeast, paving the way for biotechnological production of guaianolide-type sesquiterpene lactones.

[1] Laboratory of Plant Physiology, Wageningen University and Research, Droevendaalsesteeg 1, 6708 PB Wageningen, The Netherlands. [2] Department of Plant and Environmental Sciences, University of Copenhagen, Thorvaldsensvej 40, Frederiksberg 1871, Denmark. [3] Plant Metabolism and Metabolic Engineering Program, Centre for Research in Agricultural Genomics (CRAG) (CSIC-IRTA-UAB-UB), 08193, Barcelona, Spain. [4] Department of Biochemistry and Physiology, Faculty of Pharmacy and Food Sciences, University of Barcelona, Campus Diagonal, Av. de Joan XXIII, 27-31, 08028 Barcelona, Spain. [5] Barcelona Supercomputing Center (BSC), C/ Jordi Girona 29, 08034 Barcelona, Spain. [6] ICREA, Pg Lluís Companys 23, 08010 Barcelona, Spain. [7] Wageningen Plant Research, Wageningen University and Research, Droevendaalsesteeg 1, 6708 PB Wageningen, The Netherlands. [8] Laboratory of Organic Chemistry, Wageningen University, Stippeneng 4, 6708 WE Wageningen, The Netherlands. [9] Plant Hormone Biology group, Swammerdam Institute for Life Sciences, University of Amsterdam, Science Park 904, 1098 XH Amsterdam, The Netherlands. These authors contributed equally: Qing Liu, Arman Beyraghdar Kashkooli. Correspondence and requests for materials should be addressed to A.B.K. (email: arman.beyraghdarkashkooli@wur.nl) or to H.B. (email: h.j.bouwmeester@uva.nl)

Sesquiterpene lactones are C15 terpenoids and constitute a major class of plant secondary metabolites with diverse chemical structures. They are present in plant species in the Acanthaceae, Apiaceae and Asteraceae[1] with over 4000 different structures reported so far[2]. Sesquiterpene lactones are classified in six bicyclic or tricyclic classes named guaianolides, pseudoguaianolides, xanthanolides, eremophilanolides, eudesmanolides and germacranolides[3]. Naturally occurring sesquiterpene lactones contain an α-methylene-γ-butyrolactone or α, β-unsaturated cyclopentenone moiety. These, or more in general the the α, β-unsaturated carbonyl, largely account for their biological activity against cancer and inflammation[3,4]. Guaianolides are defined by their special 5-7-5 tricyclic structure[5] and these compounds are of interest because of their biological activity against prostate cancer[6], anti-mycobacteria activity[7], inhibition of parasite Trypanosoma cruzi growth[8] and human antioxidant response element activation at low concentrations[9]. The biosynthetic pathways of a number of important germacranolide sesquiterpene lactones have been elucidated. For instance, the biosynthesis of costunolide and parthenolide has been elucidated by isolation and characterization of the germacrene A synthase (GAS)[10,11], germacrene A oxidase (GAO)[12], costunolide synthase (COS)[13] and parthenolide synthase (PTS)[14]. In contrast, at present, enzymes involved in guaianolide formations have not been identified, preventing metabolic engineering of these classes of sesquiterpene lactones. There are some hints about the biosynthesis of guaianolides: in the family of Asteraceae and Apiaceae (e.g., feverfew (Tanacetum parthenium), chicory (Cichorium intybus), thapsia (Thapsia garganica)), two classes of sesquiterpene lactones, germacranolides and guaianolides, often occur together[4], suggesting that the biosynthesis of guaianolides may be related to that of germacranolides. Moreover, root extracts of chicory can convert costunolide to the guaianolide leucodin[2]. This bioconversion could be inhibited by a range of chemical P450 enzyme activity inhibitors[2], suggesting that one or more P450 enzymes are involved in the conversion of costunolide to leucodin[2]. On the other hand, chemical synthesis showed that one guaianolide, kauniolide, could be formed from parthenolide (Fig. 1b)[15]. In plants, parthenolide is produced from costunolide through the action of a P450 monooxygenase[14]. This implies that biosynthesis of guaianolides could proceed through parthenolide or similar epoxides[16]. However, according to Piet et al.,[17] enzyme-mediated cyclization of germacranolides could also start with allylic C3 hydroxylation of costunolide, followed by protonation of the hydroxyl group, cyclization and deprotonation which would result in the production of kauniolide, the basic guaianolide backbone (Fig. 1c). Indeed, a cytochrome P450 from feverfew encoding an enzyme capable of C3-β hydroxylation of either costunolide or parthenolide has been previously reported[14]. Neither 3β-hydroxycostunolide nor 3β-hydroxyparthenolide, however, could undergo a further cyclization towards kauniolide, suggesting that an enzyme with cyclase activity is required to catalyse this step.

Here we characterize a group of five additional P450 genes belonging to the CYP71 family that show co-expression with TpGAO, TpCOS, TpPTS, Tp3β-hydroxylase and Tp4149, which thus may act on products of these enzymes. The five genes are expressed in yeast to determine whether any of these P450 enzymes can act on costunolide or parthenolide. In these assays a gene encoding a kauniolide synthase (KLS) is identified. The encoded P450 enzyme is not only able to perform a hydroxylation on costunolide, but also, in addition, eliminates water, which is followed by cyclization and regioselective deprotonation, resulting in the product kauniolide. This enzyme thus forms the bridge between the sesquiterpene lactone class of germacranolides (costunolide) and guaianolides (kauniolide). Moreover, we show that kauniolide biosynthesis does not require a cyclase as suggested before. The full kauniolide biosynthesis pathway is successfully reconstructed in both the heterologous hosts Nicotiana benthamiana and yeast. Modelling is used to support elucidation of the putative enzymatic steps catalysed by KLS. The isolation of the KLS gene opens up opportunities for metabolic engineering of guaianolide sesquiterpene lactones.

## Results

**CYP71 activity on costunolide and parthenolide**. In feverfew extracts, multiple costunolide- and parthenolide-derived products are detected[9]. Previously, it was shown that feverfew P450s acting on germacrene A, germacrene A acid, costunolide and parthenolide all belong to the CYP71 family[11–14,18]. Moreover, the genes encoding the enzymes of the parthenolide biosynthesis pathway (TpGAS[13], TpGAO[13], TpCOS[13] and TpPTS[19]) show a distinct expression profile during feverfew ovary development, matching with the accumulation profile of parthenolide[11,13,14] (Supplementary Fig. 1a). The costunolide 3β-hydroxylase[19], though, displays a slightly different expression profile (Supplementary Fig. 1a and 1b). For our screen we therefore targeted CYP71 P450s with an expression profile similar to those of the parthenolide biosynthesis pathway genes. P450 sequences were obtained by 454 sequencing of a trichome-enriched complementary DNA (cDNA) library (www.terpmed.eu). Expression profiling was done by Illumina RNA-sequencing of six different stages of feverfew ovary development. In total, 59 P450s of the CYP71 family were detected of which 27 showed differential expression during ovary development (Supplementary Fig. 1a). Cluster analysis of these 27 candidate P450s with the parthenolide biosynthesis pathway genes resulted in a further selection of five uncharacterized P450s for which expression in ovaries shows clustering with those of the parthenolide biosynthesis pathway genes (Supplementary Fig. 1a). For four of these uncharacterized P450s (Tp9025, Tp4149, Tp8879 and Tp8886), their expression profile was independently validated by quantitative real-time reverse transcription–polymerase chain reaction (RT-qPCR), confirming the RNA-sequencing data analysis in that they display an expression profile that is similar to the bona fide pathway genes (Supplementary Fig. 1b). For those candidates not represented in the transcriptome by full-length cDNAs, RACE (rapid amplification of cDNA ends)-PCR experiments were performed in order to obtain the full sequences. Subsequently, the full-length coding sequences were cloned into yeast expression vectors for functional characterization of the encoding enzyme activity (see Methods). In this paper we describe the further characterization of four P450s: Tp8886, Tp8879, Tp4149 and Tp9025.

**Functional characterization of selected CYP71s in yeast**. Candidate P450 cDNAs were expressed in yeast WAT11[20] strain and microsomes of the transgenic yeast were isolated for in vitro testing of the enzymatic activity towards costunolide or parthenolide. Enzymes encoded by Tp4149 and Tp9025 did not show any catalytic activity on parthenolide or costunolide and their substrate specificity could not be determined. In contrast, the enzyme encoded by Tp8879 produced a product with mass $[M+H]^+ = 231.1378$ from costunolide as shown by LC-Orbitrap-FTMS (liquid chromatography-Orbitrap-Fourier transform mass spectrometry) analysis, although the peak in the chromatogram was very small. The mass reduction by 2.015 D of this product compared to the mass of costunolide ($[M+H]^+ = 233.1524$) signifies the loss of two protons which suggests introduction of a C-double bond in costunolide or a ring closure resulting in a tricyclic sesquiterpene lactone. The detection of sesquiterpene lactones with an exocyclic double bond in LC-Orbitrap-FTMS

can sometimes be improved by conjugation to glutathione or cysteine[13]. This conjugation occurs non-enzymatically, is irreversible and in case of cysteine adds an exact mass of $[M] = 121.01464$ and results in a shorter retention time (RT)[19]. The reaction product of the enzyme encoded by Tp8879 using costunolide as substrate was therefore incubated with cysteine. Analysis of the cysteine conjugates revealed two new peaks at RT = 10.07 min ($[M+H]^+ = 370.16827$) and RT = 21.41 min ($[M+H]^+ = 352.1577$). The product eluting at RT = 10.07 was identified as 3β-hydroxycostunolide-cysteine (Fig. 2d, e) according to Liu et al.[14]. More interestingly, the mass, fragmentation spectrum and RT of the product eluting at RT = 21.41 min matched that of a kauniolide-cysteine standard (Fig. 2a, b). Indeed, co-injection of kauniolide-cysteine and the enzyme products incubated with cysteine showed co-elution, showing that the enzyme encoded by Tp8879 catalyses ring closure, resulting in

kauniolide formation from costunolide (Supplementary Fig. 2). Moreover, the peak intensity of kauniolide-cysteine was 6.7 times higher than that of 3β-hydroxycostunolide-cysteine, indicating that 3β-hydroxycostunolide is a minor product (Fig. 2d, e). We therefore called Tp8879 kauniolide synthase (TpKLS). When microsomes expressing *TpKLS* were incubated with parthenolide, new peaks were not detected, not even after incubation of the enzyme products with cysteine. This suggests that costunolide is the natural substrate for TpKLS and that TpKLS is the first committed enzyme in the biosynthesis towards guaianolides, revealing that guaianolide-type sesquiterpene lactones (like kauniolide) are derived from germacranolide-type sesquiterpene lactone (i.e., costunolide).

The production of 3β-hydroxycostunolide, though as a minor product, from feeding costunolide to TpKLS suggests that this enzyme can perform hydroxylation of costunolide at the C3

**Fig. 1** Biosynthetic pathway of costunolide and proposed (bio)synthetic pathways for kauniolide. **a** Biosynthetic pathway of costunolide[13]. **b** Conversion of costunolide to parthenolide by PTS[14] and chemical biomimetic synthesis of kauniolide from parthenolide[15]. **c** Proposed biosynthesis of kauniolide[17] and proposed biosynthetic pathway of leucodin in chicory[2]. **d** Biosynthetic pathway of kauniolide, through 3α-hydroxycostunolide as intermediate, as identified in the present study. GAS germacrene A synthase, GAO germacrene A oxidase, COS costunolide synthase, PTS parthenolide synthase, p-TSA p-toluene sulphonic acid, $CH_2Cl_2$ dichloromethane, RT room temperature, $POCl_3$ phosphoryl chloride

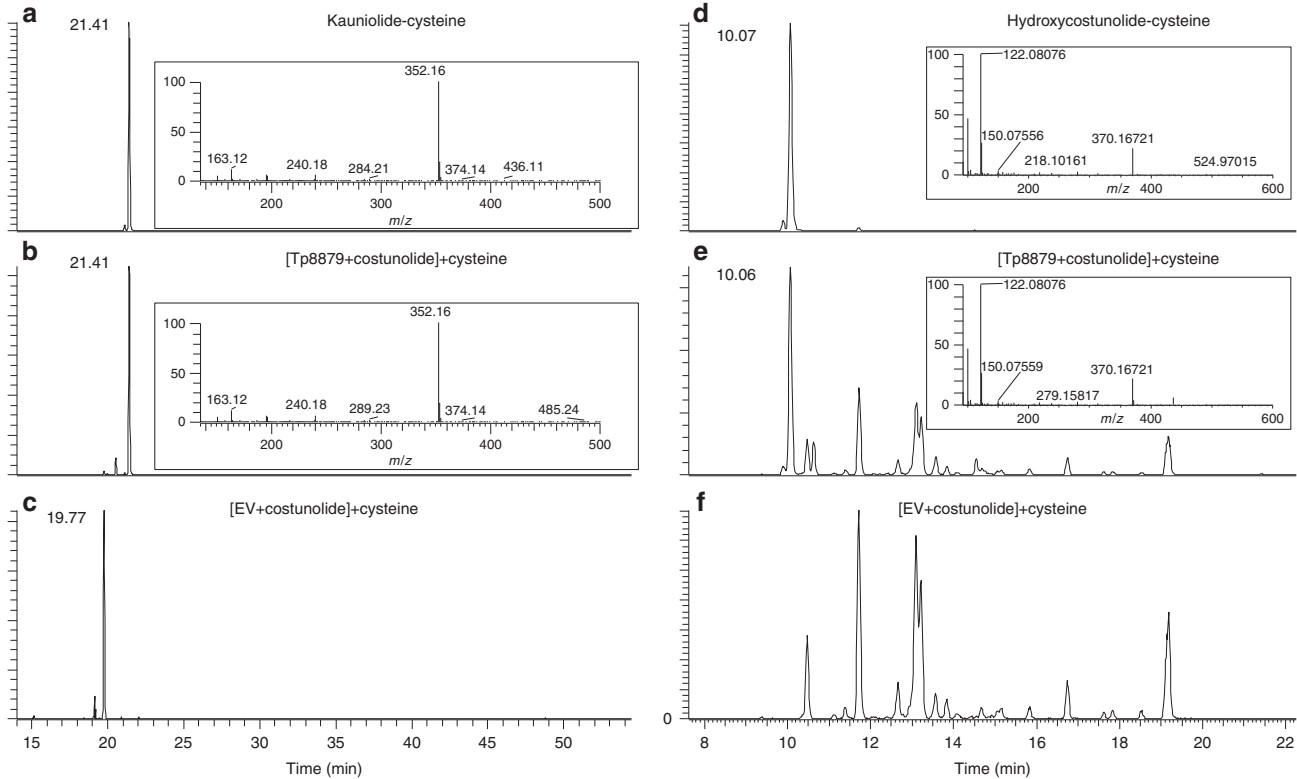

**Fig. 2** Identification of kauniolide-cysteine in an assay with yeast microsomes using LC-Orbitrap-FTMS. **a** Incubation of kauniolide with cysteine. **b** Microsomes expressing Tp8879 (TpKLS) fed with costunolide and incubated with cysteine. **c** Negative control where no kauniolide-cysteine was formed by feeding costunolide to microsomes expressing an empty vector (EV) and incubation with cysteine. **d** Formation of 3β-hydroxycostunolide-cysteine by incubating 3β-hydroxycostunolide with cysteine. **e** LC-Orbitrap-FTMS chromatograms at $m/z = 370.16827$ (10 ppm, positive ionization mode, mass for 3β-hydroxycostunolide-cysteine) of 3β-hydroxycostunolide-cysteine formed in the enzymatic assay of feeding costunolide to TpKLS and incubated with cysteine. **f** Negative control where no 3β-hydroxycostunolide-cysteine was formed by feeding costunolide to EV (microsomes of yeast with empty vector) and incubation with cysteine

position. Moreover, 3-hydroxycostunolide has been suggested to be an intermediate in kauniolide biosynthesis[17]. We have previously identified a P450 in feverfew which can hydroxylate costunolide at C3 (Tp3β-hydroxylase[19]), but does not produce kauniolide. To assess if 3β-hydroxycostunolide is an intermediate in the production of kauniolide we incubated TpKLS with 3β-hydroxycostunolide. This did not result in conversion into kauniolide, indicating that 3β-hydroxycostunolide cannot serve as a substrate for TpKLS (Supplementary Fig. 3). The product of the enzyme encoded by Tp8886 enzyme could not be identified due to its low abundance.

**In silico docking of costunolide in TpKLS**. To get further insight into the mechanism by which kauniolide is formed by TpKLS, we used in silico modelling of TpKLS, followed by costunolide substrate docking experiments. As validation for the modelling approach we used two other P450 enzymes that use costunolide as substrate and for which the site of costunolide oxidation is known, i.e., epoxidation of C4–C5 by TpPTS and hydroxylation of C3 by costunolide 3β-hydroxylase[13,14]. The protein structure models of TpPTS, costunolide 3β-hydroxylase and TpKLS were generated using the crystal structure of *Homo sapiens* P450 2C9[20] (Protein Data Bank (PDB) code 4GQS), *Oryctolagus cuniculus* P450 2C5[21] (PDB code 1NR6) and *Homo sapiens* dual specificity protein phosphatase 13 (PDB code 2PQ5_A), respectively (Supplementary Table 2). Substrate docking of costunolide was simulated in PELE (Protein Energy Landscape Exploration) for each of the

three modelled enzyme structures[22,23]. Two models were used: (i) a free simulation where the substrate has no constraints, and (ii) simulations with a defined constraint, where the substrate was not allowed to move beyond 15 Å from the haem$^+$-Fe(IV)-O$^{2-}$ (noted as haem oxyanion). Following a Monte Carlo (MC) move where the substrate and the protein are first perturbed and then relaxed, the substrate binding in PELE is scored by the enzyme–substrate interaction energy, as described by the OPLS (optimized potential for liquid simulations) force field. PELE was ran on each system for maximally 5000 MC steps, after which the distance from the haem oxyanion to the sesquiterpene lactone and docking orientation was scored, thereby performing a population analysis (Fig. 3b, c).

The in silico docking studies of costunolide with TpKLS indicate that the C3 position of costunolide is highly favoured to interact with the haem oxyanion in the TpKLS model structure (Fig. 3b), for both the constrained and unconstrained models (1379 and 1417 occasions, respectively). The calculated distance of costunolide C3 to the haem oxyanion in the model is 3.6 Å which translates into a hydrogen atom abstraction coordinate ~2.2 Å, sufficient to trigger hydroxylation at costunolide C3. Also, in the in silico docking experiments of costunolide with costunolide-3β-hydroxylase, the PELE simulations indicated a preferred regioselective C3 orientation of costunolide towards the haem oxyanion which is in agreement with its enzymatic activity (Fig. 3b, d; Supplementary Fig. 4). Moreover, the in silico docking of costunolide in TpKLS indicates that α-hydroxylation of C3 is favoured (Fig. 3d; Supplementary Fig. 4), while for costunolide-

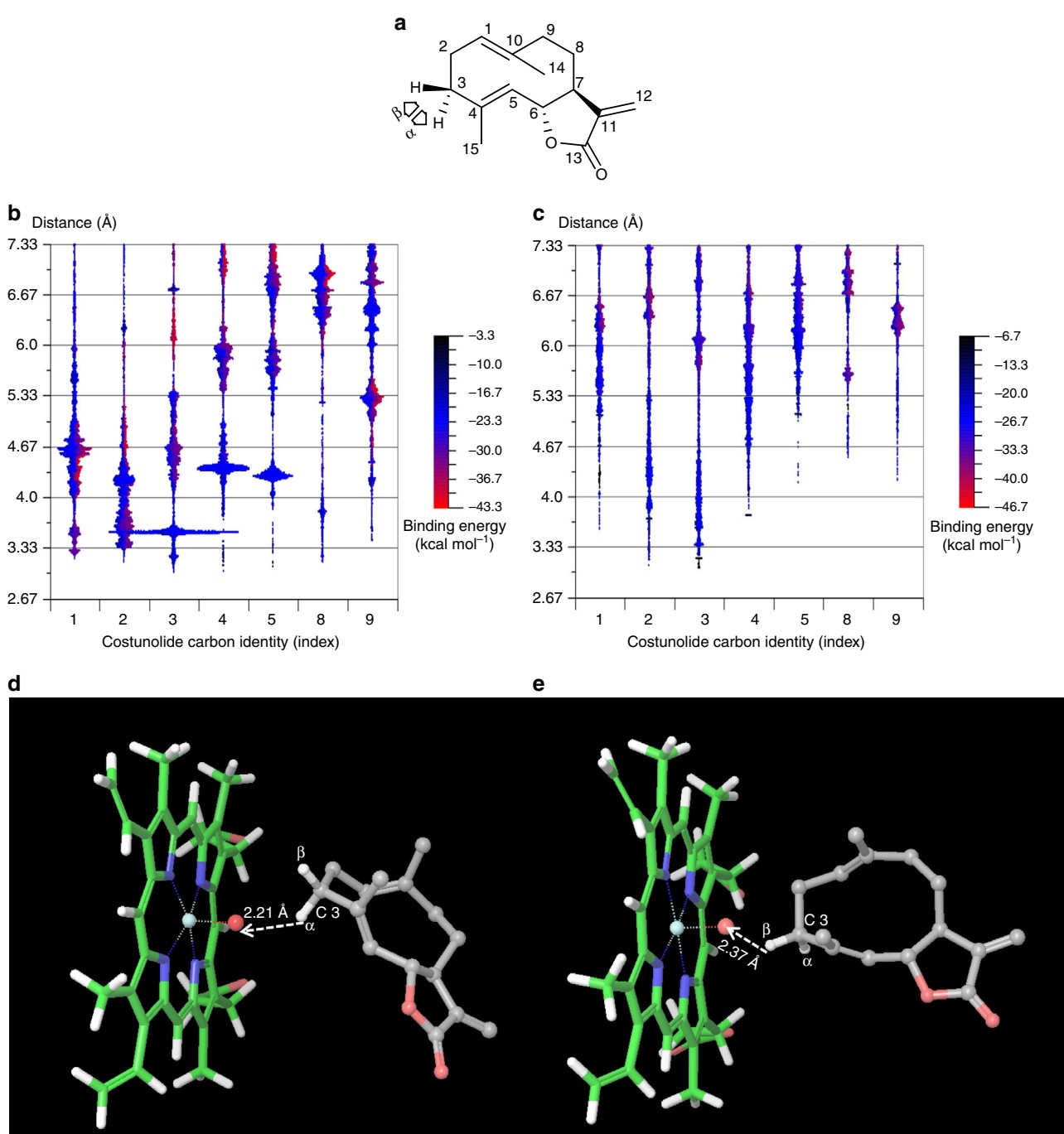

**Fig. 3** In silico docking of costunolide into the active site of the feverfew enzymes, kauniolide synthase and costunolide 3β-hydroxylase. **a** Structure of costunolide and pro-α and pro-β hydroxylation orientations. **b** Preferred docking orientations of costunolide as calculated using Protein Energy Landscape Exploration software. Costunolide carbon distance distribution relative to the haem oxyanion in a KLS homology model (constrained model, at 15 Å). **c** Costunolide carbon distance distribution relative to the haem oxyanion in a costunolide 3β-hydroxylase homology model (constrained model, at 15 Å). In silico docking of costunolide into the active site of kauniolide synthase and costunolide 3β-hydroxylase by Glide. **d** Preferred docking of costunolide in the active site of kauniolide synthase is in α-orientation. **e** Preferred docking of costunolide in the active site of costunolide 3β-hydroxylase is in β-orientation. Colour keys of **b**, **c** represent the binding energy (kcal mol⁻¹) expressed in negative values

3β-hydroxylase, in agreement with its enzymatic function, β-hydroxylation of C3 is preferred (Fig. 3c, e). Presumably, α-hydroxylation of costunolide at C3 is directly followed by protonation, dehydration, cyclization and deprotonation in the enzymatic cavity, resulting in the formation of kauniolide. In contrast, the less preferred β-hydroxylation—which seems to occur, though at a much lower rate—at costunolide C3 apparently

cannot be protonated and is removed from the enzymatic cavity as a minor side product.

Although for these two enzymes the predictions from the in silico modelling match with the actual enzyme activity, for TpPTS the modelling suggests epoxidation of C9 while the enzyme epoxidizes C4–C5 of costunolide (Supplementary Fig. 5). The number of predictions for this preferred costunolide C9

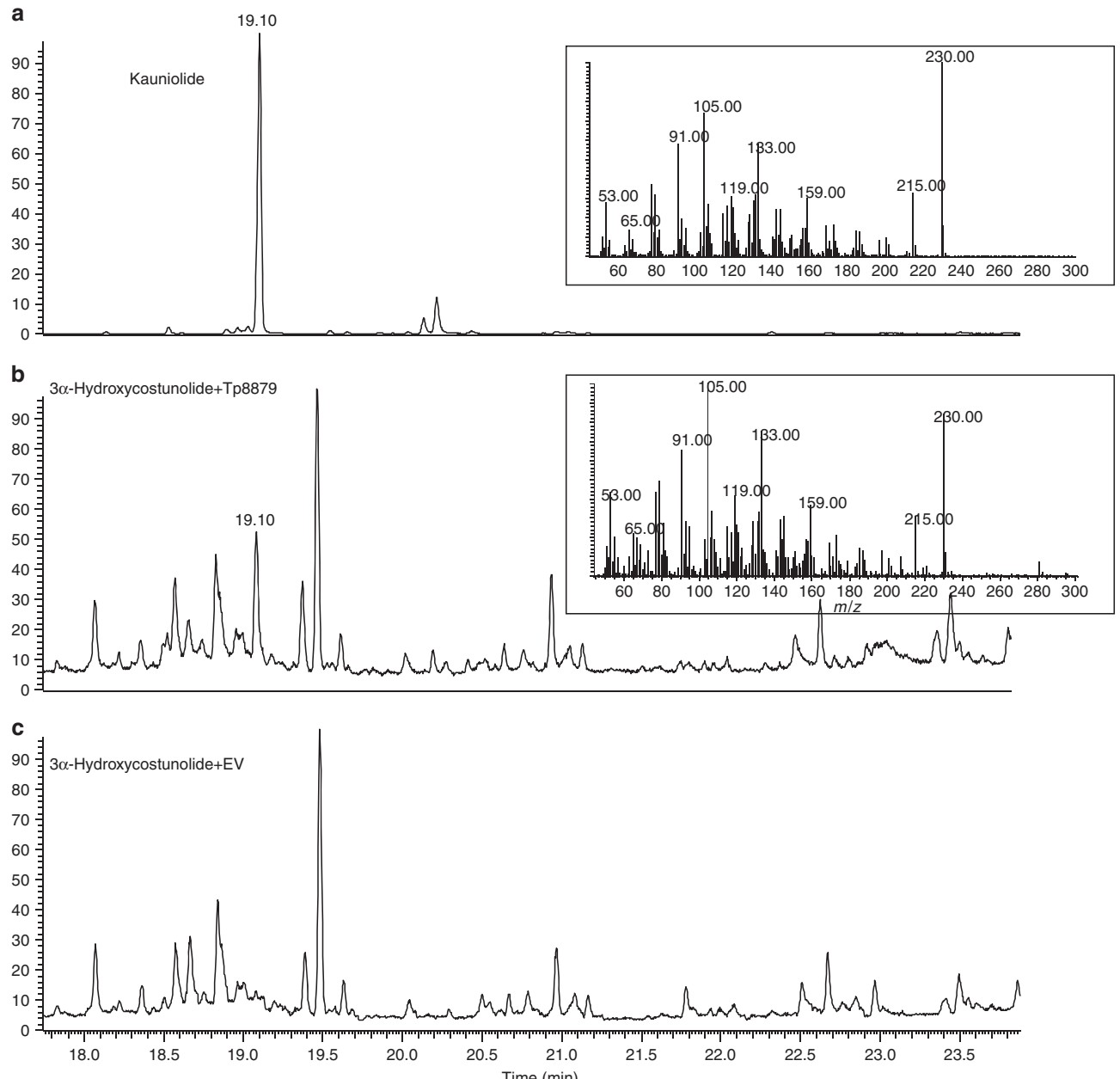

**Fig. 4** The 3α-hydroxycostunolide serves as the intermediate compound in kauniolide formation. **a** GC-MS chromatogram of kauniolide standard. **b** GC-MS chromatogram of product of assay with 3α-hydroxycostunolide as substrate fed to yeast microsomes expressing *Tp*8879: production of kauniolide (nominal mass 230). **c** Negative control; feeding 3α-hydroxycostunolide to microsomes containing EV (microsomes of yeast with empty vector) does not lead to kauniolide formation

orientation towards the haem of *TpTP*S was much lower (~252) than the number of preferred orientations of costunolide C3 towards the haem of TpKLS (1417) and costunolide 3β-hydroxylase (365), suggesting that the model of costunolide docking to TpPTS is less reliable than for the other two enzymes.

**3α-Hydroxycostunolide is intermediate substrate of TpKLS.** We therefore, according to in silico docking studies, hypothesized that 3α-hydroxycostunolide serves as the intermediate substrate in kauniolide biosynthesis. Hence, we chemically synthesized 3α-hydroxycostunolide (analysed by gas chromatography–mass spectrometry (GC-MS); Supplementary Fig. 6) and fed it to TpKLS. GC-MS analysis of extracts from feeding experiments

showed conversion of 3α-hydroxycostunolide to kauniolide by TpKLS (Fig. 4a, b). No kauniolide was detected in samples where 3α-hydroxycostunolide was incubated with EV (microsomes of yeast with empty vector) (Fig. 4c). We therefore suggest that conversion of costunolide into kauniolide starts with hydroxylation at the C3 position, predominantly in α-orientation and with β-orientation as a minor side reaction. This is immediately followed by a ring closure reaction for 3α-hydroxycostunolide, which results in kauniolide formation, while the other intermediate, 3β-hydroxycostunolide, leaves the enzymatic cavity as side product. The occasional release of the intermediate 3β-hydroxycostunolide from the enzymatic cavity would explain the presence of 3β-hydroxycostunolide as a side product of TpKLS.

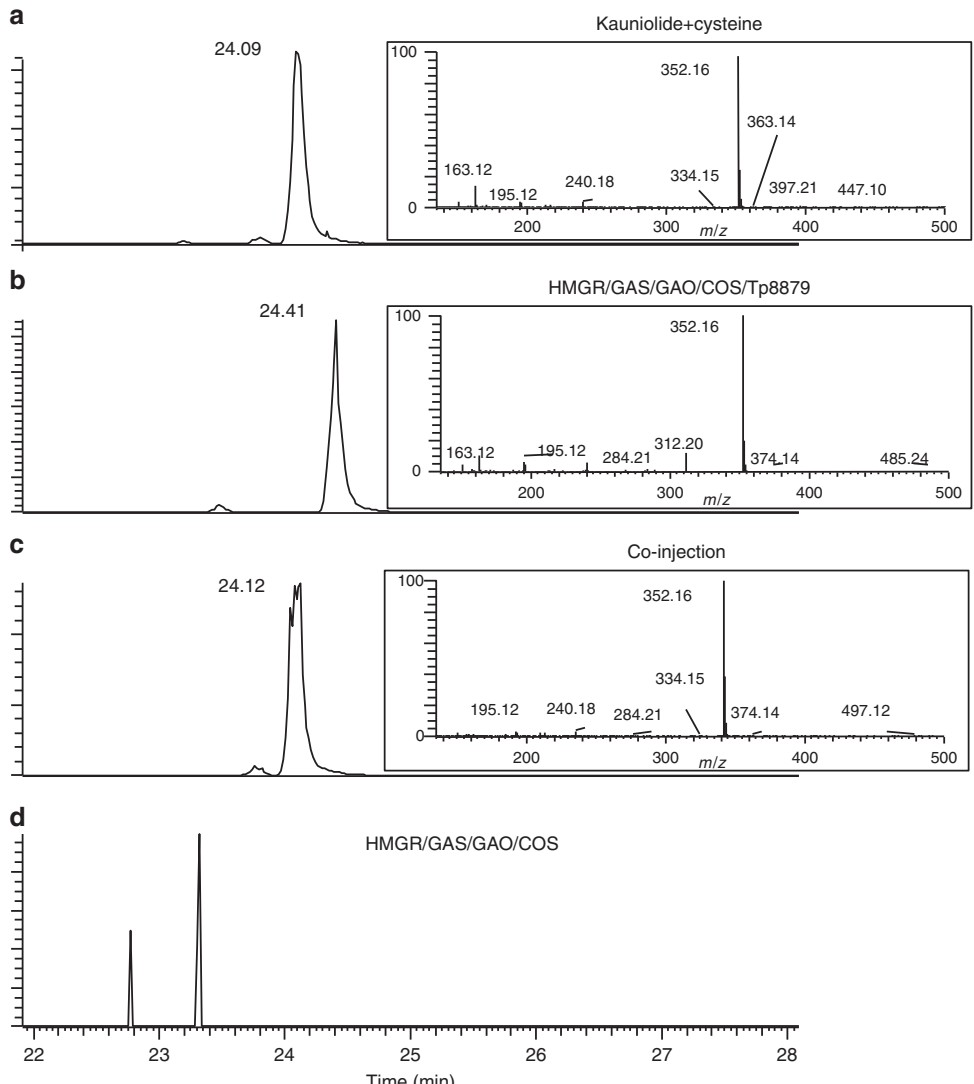

**Fig. 5** Characterization of TpKLS and reconstitution of kauniolide biosynthesis pathway in *N. benthamiana*. LC-Orbitrap-FTMS chromatograms at [M+H]$^+$ = 352.15771 of kauniolide-cysteine. **a** Kauniolide-cysteine standard formed from incubation of kauniolide and cysteine. **b** Kauniolide-cysteine production by expressing the costunolide biosynthetic pathway (AtHMGR, TpGAS, TpGAO and CiCOS) in combination with Tp8879 (TpKLS). **c** The chromatogram of co-injection of kauniolide-cysteine and the product of the costunolide biosynthetic pathway in combination with Tp8879 (TpKLS). **d** Negative control. Expression of the costunolide biosynthetic pathway does not lead to formation of kauniolide-cysteine

**In planta reconstruction of kauniolide biosynthetic pathway.** Previously, we reconstituted the full biosynthesis pathway towards costunolide and parthenolide by transient gene expression in *N. benthamiana*[13,14]. To obtain production of kauniolide in *N. benthamiana*, the *TpKLS* was cloned into a binary plant expression vector under control of the Rubisco small subunit promoter (RBC) which was used for agrobacterium transformation. All genes of the biosynthetic pathway towards kauniolide (*TpGAS*, *CiGAO*, *CiCOS* and *TpKLS*) were subsequently transiently expressed in *N. benthamiana* leaves by co-agro-infiltration. In addition, overexpression of the *Arabidopsis thaliana* 3-hydroxy-3-methylglutaryl-CoA reductase (*AtHMGR*) was used to boost farnesyl-diphosphate production, which is the precursor of all sesquiterpenoids. *N. benthamiana* leaves were harvested 4 days post agro-infiltration (4 dpi), extracted with methanol/formic acid (0.1%) and extracts were measured by LC-Orbitrap-FTMS using targeted analysis for free and conjugated products (costunolide, kauniolide and costunolide/kauniolide-cysteine, and costunolide/kauniolide-glutathione conjugates). No free kauniolide was detected, but both costunolide-cysteine and a

mass presumed to be kauniolide-cys ([M+H]$^+$ = 352.1577) were detected in the leaf extracts (Fig. 5b), indicating that kauniolide produced in *N. benthamiana* leaves is conjugated to cysteine.

To confirm the identity of the compound with mass 352.1577, kauniolide was (non-enzymatically) conjugated in vitro to cysteine to produce kauniolide-cys. Indeed, retention time and molecular mass ([M+H]$^+$ = 352.1577) of kauniolide-cys were identical to the product obtained in planta upon expression of the kauniolide biosynthetic pathway (Fig. 5a–c). Neither kauniolide-cysteine nor kauniolide-glutathione were detected in control samples (costunolide pathway co-expressed with empty vector).

**Reconstruction of kauniolide biosynthetic pathway in yeast.** Recently, reconstruction of the biosynthetic pathways of the sesquiterpene lactones, costunolide and artemisinin, in yeast was demonstrated[24,25]. To achieve the same for kauniolide, we cloned the four biosynthetic genes (*TpGAS*, *TpGAO*, *TpCOS* and *TpKLS*) into two dual yeast expression vectors and transformed yeast cells by plasmid transformation under the control of a galactose-

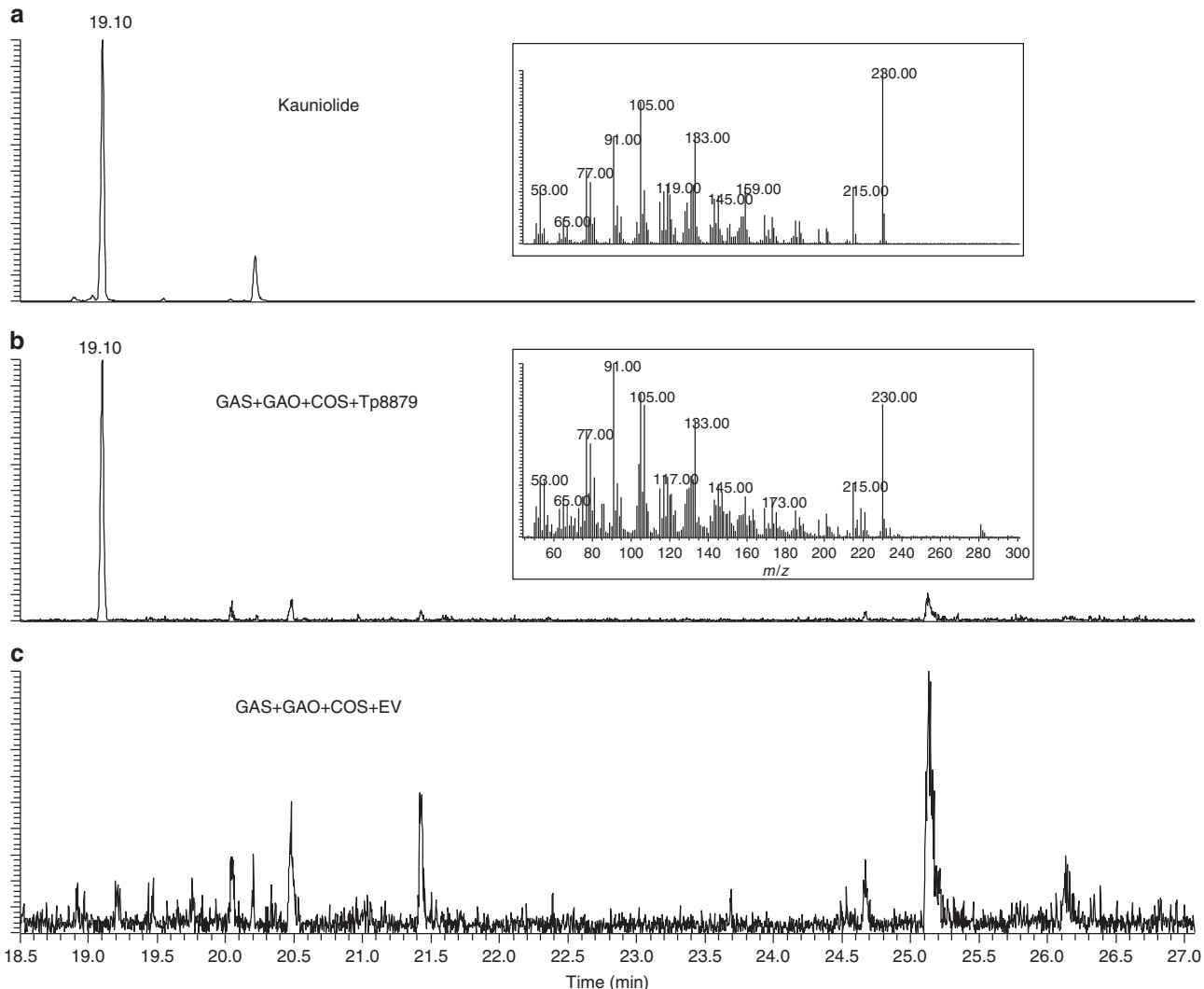

**Fig. 6** Characterization of TpKLS and reconstitution of the kauniolide biosynthesis pathway in yeast. GC-MS chromatograms at nominal mass 230 of kauniolide. **a** Kauniolide standard. **b** Kauniolide production by expressing the costunolide biosynthetic pathway (TpGAS, TpGAO and CiCOS) in combination with Tp8879 (TpKLS). **c** Negative control. Expression of the costunolide biosynthetic pathway with EV (microsomes of yeast with empty vector) does not lead to formation of kauniolide

inducible promoter. Yeast cultures were induced by galactose and after 72 h cells plus medium were extracted with ethyl acetate. The extracts were concentrated, dried over a $Na_2SO_4$ column and the dehydrated sample was injected into GC-MS for analysis. Yeast cells expressing TpGAS, TpGAO and TpCOS produced costunolide, confirming previous results[20]. The GC-MS chromatogram from the yeast cells expressing *TpGAS*, *TpGAO*, *TpCOS* and *TpKLS* showed a new peak at 19.10 min, with the same retention time and mass spectrum as the kauniolide standard (Fig. 6a, b). This indicates a successful production of free kauniolide in yeast cells. No conjugated kauniolide/costunolide was detected by GC-MS.

## Discussion

Several sesquiterpene lactones have been identified in feverfew[9] and several biosynthetic pathway enzymes and their corresponding genes have been characterized such as germacrene A synthase, costunolide synthase, parthenolide synthase and costunolide 3β-hydroxylase[13,19]. Nevertheless, the biosynthesis of several feverfew sesquiterpene lactones (e.g., santamarine, reynosin and artecanin) is still a mystery. Costunolide may play a central role in sesquiterpene lactone biosynthesis since several

P450s have been shown to use this compound as substrate for oxidation[18,19]. Furthermore, other researchers have proposed that costunolide might be the branching point for biosynthesis of other classes of sesquiterpene lactones such as guaianolides and eudesmanolides[2,18,26].

Here, we demonstrate the catalytic activity of a CYP71 (TpKLS) responsible for the production of a guaianolide-type sesquiterpene lactone, kauniolide, from a germacranolide-type sesquiterpene lactone, costunolide. Guaianolide biosynthesis has been suggested to start with C3 hydroxylation of costunolide, followed by cyclase activity by a separate enzyme[17]. The cyclization would involve the following steps: (i) protonation of the newly introduced hydroxyl group; (ii) loss of water, producing an allylic carbocation $C3^+–C4 = C5/C3 = C4–C5^+$; (iii) nucleophilic attack of the C1–C10 double bond to the C5 carbocation, forming the annelated 5–7 bicyclic system and a tertiary carbocation at C10; and (iv) regioselective deprotonation at C1 giving a C1–C10 double bond. It is likely that at least some of these steps occur in a concerted fashion[17]. Here we show that all these reaction steps take place within a single P450. The production of 3β-hydroxycostunolide as a minor side product of TpKLS enzyme activity confirms that the enzyme hydroxylates costunolide at the

**Fig. 7** Proposed mechanism of action of TpKLS. Costunolide is hydroxylated at the C3 position (in which the α-orientation is favoured). We postulate that C3 α-hydroxylation is followed by protonation and water abstraction resulting in an allylic carbocation. The C5 carbocation resonance form is attacked by the C1–C10 double bond which results in C1–C5 ring closure. Subsequent deprotonation results in kauniolide. Note that proposed protonation has not been proven in this study

C3 position. It has been shown that a different CYP71 from feverfew (costunolide 3β-hydroxylase), stereoselectively adds a hydroxyl group in β-orientation to the C3 position of costunolide but does not convert it to kauniolide[14]. Using yeast microsome feeding assays, we show that 3β-hydroxycostunolide cannot be converted to kauniolide by TpKLS. Indeed, protein modelling suggests that C3-β-hydroxylation of costunolide does not allow the correct orientation in the active site of TpKLS for further postulated protonation (Fig. 3d). In contrast, α-orientation of the C3-hydroxyl group results in a suitable orientation of the substrate in the enzyme cavity which allows such a reaction to occur. Costunolide was indeed docked in a rather peculiar way in TpKLS. A conserved active site threonine in α-helix I appears to have different functions for different substrates in literature[27,28] and may have an additional role in this system. It may well be that the conserved threonine of α-helix I protonates costunolide after the hydroxylation step. Protonation by this threonine does not occur in the other discussed enzymes. After ring closure the conformation of the product is very different from the substrate, enabling it to move out of the active site, after which the threonine can be protonated again. Perhaps the threonine oxyanion is involved in the regioselective deprotonation of the intermediate carbocation, leading to kauniolide. We therefore suggest that TpKLS has costunolide C3-hydroxylase activity with a preference for α- over the β-orientation, while the 3α-hydroxycostunolide is subsequently converted to kauniolide by the same enzyme. We note that modelling results are based on a heterologous P450 template model. Future acquisition of the TpKLS crystal structures may substantially improve the in silico simulation.

Other enzymes in the feverfew sesquiterpene lactone biosynthesis pathway also display a number of sequential activities, all initiated by oxidation. For instance, the P450 TpGAO converts germacrene A into germacrene A alcohol, germacrene A aldehyde and finally germacrene A acid[13]. A similar type of single-enzyme

multiple reactions also occurs in the closely related *Artemisia annua*, in which amorphadiene is converted by CYP71AV1 into amorphadiene alcohol, amorphadiene aldehyde and finally artemisinic acid[29]. Also, the hydroxylation of germacrene A acid by COS is followed by spontaneous lactone formation[2,13]. However, in all these examples the sequential enzyme activities target the same position in the substrate. What makes TpKLS unique is that this enzyme not only carries out the conventional P450 catalysed hydroxylation, but also the protonation and dehydration, cyclization and subsequent deprotonation (Fig. 7).

Phylogenetic analysis of the cytochrome P450s from feverfew and some other Asteracae plants shows that TpKLS clusters more closely with COS proteins than with PTS or costunolide 3β-hydroxylase and the GAOs (Supplementary Fig. 8). Analysis of more than 2.3 million expressed sequenced tags (ESTs) from different Asteracae species (http://compgenomics.ucdavis.edu) identified contigs from these species having significant homology with TpKLS (but still with less than 65% homology) (Supplementary Fig. 7, Supplementary Table 1). Indeed, some of these Asteracae species are known to produce guaianolide-type sesquiterpene lactones such as *Cichorium* spp.[30] (lactucin, lactucopicrin and 11β,13-dihydrolactucin), *Cynara cardunculus*[31] (aguerin B, cynaropicrin). This suggests that a specific and quite unique protein with KLS activity has evolved in feverfew out of COS, but that putative KLSs in other Asteracae may have evolved independently. This is in stark contrast with the strong sequence similarity of COS genes in Asteracae species, resulting in a strong clustering in phylogenetic analysis (Supplementary Fig. 8).

Feverfew indeed contains guaianolide sesquiterpene lactones, artecanin and tanaparthin-β-peroxide[9]. Artecanin has antiproliferative properties on leukaemia cancer cells[32]. Artecanin biosynthesis requires multiple oxidation steps of kauniolide, likely catalysed by P450s, and these may be identified by feverfew P450

expression profiling and comparison with the accumulation pattern of artecanin (Supplementary Fig. 9). Artecanin shows a stable accumulation level during development of feverfew flowers, but suddenly increases late during development of the flowers. The increased production of artecanin in this late stage of flower development may be due to reduced expression of PTS[14] and therefore reduced competition with PTS for the costunolide substrate. Our elucidation of the biosynthesis of kauniolide, the most basic guaianolide, will provide opportunities for bio-engineering of valuable guaianolides like artecanin.

## Methods

**RNA-sequencing and expression analysis.** The *T. parthenium* EST library used in the present study was made from flower messenger RNA as previously reported [11]. Later Illumina reads of ovary developmental stages were mapped[33] against this assembly and eXpress (https://pachterlab.github.io/eXpress/) was accommodated for expression level estimation in different developmental stages. DEseq[34] was used to check differential expression levels for three biological replicates. A negative binomial distribution test with the false discovery rate threshold of 0.05 was used for DEseq. Mean expression values of CYP71 candidates together with feverfew sesquiterpene lactone biosynthesis pathway genes were directly imported to the software GeneMaths XT (2.12). This was followed by log transformation and normalization of the complete set with the average algorithm for data offset calculation. A complete linkage algorithm was used for cluster analysis (Fig. 2a). Focussing on genes showing a similar expression pattern as the pathway genes limited the number of candidates to six, which were clustering together with other known biosynthesis pathway genes.

**Gene expression analysis.** Relative gene expression of candidate P450s was measured by quantitative real-time RT-PCR. Total RNA and cDNAs were obtained from ovary developmental stages. Ovaries were separated from the flowers. Approximately 3 g of ovaries were ground in a Greiner tube with 10 mL liquid $N_2$. Tripure (Roche, Mannheim, Germany) isolation reagent was used for total RNA extraction. The isolated RNA was then treated with DNAse I (Invitrogen, USA) and purified using the RNeasy RNA clean-up kit (Qiagen, USA). The reverse transcription was performed using the Taqman Reverse Transcription reagent kit. Real-time RT-PCR was performed using a LightCycler 480 (Roche Diagnostics). The LightCycler experimental run protocol used was: 95 °C for 10 min, 95 °C for 10 s, 60 °C for 30 s for 40 cycles and finally a cooling step to 40 °C. LightCycler Software 1.5.0 was used for data analysis. In order to determine the efficiency, a six serial dilution series point (from 200 to 6.25 ng) was used which was performed in triplicate. Primer pairs for *TpGAS* and *TpActin* were forward actin 5′-CCTCTTAATCCTAAGGCTAATC-3′: reverse actin 5′-CCAGGAATCCAGCACAATACC-3′; forward TpGAS 5′-TTCTCCTCTTATTCTCAACTGTGG-3′; reverse TpGAS 5′-TGCTATCTCGGGTACTTTCAAGG-3′. The following primer pairs were used for *TpGAO*, *TpCOS* and *TpPTS* amplification: forward *TpGAO* 5′-TGCAGCTCCCGCTTGCTAATATAC-3′, reverse *TpGAO* 5′-AGTCTTTCTTTGAACCGTGGCTCC-3′, forward *TpCOS* 5′-TAGCTTCATCCCGGAGCGATTTGA-3′, reverse *TpCOS* 5′-AAATTCTTCGGCCCGCACCAAATG-3′, forward *TpPTS* 5′-AGACATTACGTTTACACCCTCCCG-3′, reverse *TpPTS* 5′-ATCACGACACAAGTCCCAGGGAAA-3′, forward Tp8879 5′-AGTTCCTTACGGCCATTTCTGGGA-3′, reverse 5′-AGAAGATCGTCCTCAGTTGCTCCA-3′. Transcript levels were quantified in three independent biological replicates and three technical replicates for each biological replicate. ΔCT was calculated according to the following formula: ΔCT = CT (Target)−CT (Actin). Fold change calculation was done using the $2^{-\Delta CT}$ equation [35].

**Isolation and cloning of candidate genes from feverfew.** Six candidate cytochrome P450 contigs were identified by sequence homology to known sesquiterpene monooxygenases. These candidates were selected for functional characterization. A RACE-PCR (Clontech) approach was used to obtain the complete coding sequence of the 5′- and/or 3′-regions. The coding sequence (CDS) was amplified by PCR and at the same time introducing *NotI* and *PacI* restriction sites. The CDS was subsequently cloned into a yeast expression vector, pYEDP60. Both the TPS and P450 cDNA sequences used in this study were identified in the same glandular trichome-enriched cDNA library and we assume that both genes are expressed in the same cell. The cDNA sequence of the candidate genes has been submitted in GeneBank under the accession numbers MF197558 (Tp8879) and MF197559 (Tp8886). The sequences were also deposited in David Nelson's cytochrome P450 database (http://drnelson.uthsc.edu/cytochromeP450.html); Tp8879 and Tp8886 were assigned the names CYP71BZ6X and CYP71DD5, respectively[36].

**Plasmid construction for gene expression in yeast.** Candidate P450s were cloned into the pYED60 vector using *NotI/PacI* restriction sites. The obtained constructs were transformed into the WAT11[37] yeast strain. After transformation yeast clones were selected on synthetic dextrose (SD) minimal medium supplemented with amino acids, but omitting uracil and adenine sulphate for auxotrophic selection of transformants. For reconstruction of the kauniolide biosynthesis pathway in yeast, we used the *CiGAO*

and *TpGAS* genes which were previously cloned into the pESC-Trp yeast expression vector (Agilent Technologies)[13]. CiCOS was subcloned from pYEDP60[13] using digestion by *NotI* and *PacI* and inserted into pESC-Ura. Subsequently, TpKLS was cloned by addition of *BamHI* and *KpnI* sites into CiCOS-pESC-Ura which resulted in CiCOS+TpKLS-pESC-Ura. WAT11 yeast[37] was transformed with CiGAO+TpGAS-pESC-Trp and CiCOS+TpKLS-pESC-Ura and colonies were selected on SD minimal medium (containing 0.67% Difco yeast nitrogen base medium without amino acids, 2% glucose and 2% purified agar) which was supplemented by amino acids but omitting L-tryptophan and uracil for auxotrophic selection of transformants. The combination of CiGAO+TpGAS-pESC-Trp and CiCOS-pESC-Ura with empty multiple cloning site 2 (MSC2) was used as control represented by GAS+GAO+COS+EV (empty vector). Then, 2000 mL of each yeast culture (medium and cells) was extracted twice with 500 mL of ethyl acetate. Extracts were concentrated to 100 μL using a rotary evaporator and a $N_2$ flow and used for GC-MS analysis.

**Yeast microsome isolation and in vitro microsome assay.** Microsome isolation of transformed yeasts was done according to ref. [38] with some modifications. Transformed yeast single colonies were pre-cultured in 50 mL SGI medium for 3 days at 30 °C and shaking at 300 rpm. Then, a 250 mL YPL medium containing 2% galactose was added for induction of gene expression and was kept for 24 h under the same conditions. Subsequently, the cells were harvested and chilled on ice for 20 min, followed by centrifugation at 4900 × g for 10 min. Cell pellets were resuspended in 100 mL extraction buffer (50 mM Tris-HCl pH 7.5, 1 mM EDTA, 0.6 M sorbitol and 10 mM β-mercaptoethanol) and kept for 10 min at room temperature. Again, the cells were centrifuged (4900 × g for 10 min) and cells were washed three times (3 mL) with extraction buffer, but omitting β-mercaptoethanol. The centrifuge tube was rinsed with 2 mL of the same buffer and the mixture was transferred to a 50 mL Falcon tube. Approximately 25 mL of glass beads (450 μm) were added to the tubes and cells were lysed by vigorous shaking in a cold room for 10 min. The cell lysate was transferred to a 25 mL centrifuge tube and were centrifuged at 10,500 × g for 10 min. Later, the supernatant was again centrifuged at 195,000 × g for 2 h. The pellets (microsomal fractions) were resuspended in a 4 mL solution of ice-chilled 50 mM Tris-HCl (pH 7.5) containing 1 mM EDTA and 20% (v:v) glycerol using a glass Tenbroek homogenizer. Aliquots of these microsomal fractions were kept at −80 °C until use.

Yeast microsome in vitro assays were done in a mixture of 28.8 μl isolated microsomal fractions, 4 μl substrate (10 mM in DMSO), 40 μl NADPH (10 mM in 100 mM potassium phosphate buffer), 8 μl potassium phosphate buffer (1 M, pH 7.5), and 90.4 μl of water which was incubated for 2.5 h at 25 °C with shaking (200 rpm). The obtained mixture was centrifuged at 12,000 × g (4 °C) and supernatant was passed through a 0.22 μm filter before injection into LC-MS. Yeast microsome in vitro assays with 3α-hydroxycostunolide as substrate were performed as described above, but after 2.5 h of incubation at 25 °C and shaking at 200 rpm, samples were extracted 3 times with 2 mL ethyl acetate. The ethyl acetate phase was passed through a pasteur pipet, plugged with glass wool and filled with $Na_2SO_4$ to dehydrate the extract. The extract was then concentrated and injected into GC-MS.

**Plasmid construction for expression in *N. benthamiana*.** Heterologous transient expression of candidate P450s of *N. benthamiana* was done according to ref. [14]. Briefly, candidate P450 CDSs were cloned into Impactvector 1.1 under the control of Rubisco (RBC) promoter[14]. Other pathway genes (*GAO*, *COS*, *PTS*) were also cloned into the same expression vector. *TpGAS* was also cloned into ImpactVector1.5 to fuse it with the RBC promoter and the CoxIV mitochondrial targeting sequence as it was shown before that mitochondrial targeting of sesquiterpene synthases results in improved sesquiterpene production[13]. Later, the CDS was transformed into the pBinPlus binary vector[39] by an LR reaction (Gateway-LR Clonase TM II) to put the CDS between the right and left borders of transfer DNA (T-DNA) for plant transformation. All these constructs were finally transformed into an AGL-0 *Agrobacterium* strain. An LR reaction (Gateway-LR Clonase TM II) was carried out to clone each gene into the pBinPlus binary vector[39] between the right and left borders of the T-DNA for plant transformation.

**Transient expression in *N. benthamiana*.** Transient heterologous expression of biosynthetic pathway genes was done by *Agrobacterium*-mediated transformation (agro-infiltration) of *N. benthamiana* plants. Transformed *Agrobacteria* were grown at 28 °C at 250 rpm for 48 h in LB media with proper antibiotics. Then, *Agrobacterium* cells were harvested at 4000 × g for 20 min at room temperature. Cells were then resuspended in agro-infiltration buffer (10 mM MES, 10 mM $MgCl_2$ and 100 μM acetosyringone). The final optical density (OD) in the 600 nm wavelength absorbance was set to 0.5. Gene mixture dosage for control treatments were adjusted by addition of representative number of empty vector(s). Cultures were mixed on a roller-mixer for 2.5 h (50 rpm). The 4-week-old *N.benthamiana* plants were agroinfiltrated with a needleless 1 mL syringe injection to the abaxial side of the leaf. Transformed plants were kept for another four and half days and then harvested for metabolites analysis.

**LC-MS analysis of yeast microsome assay and leaf extracts.** In order to analyse the products formed in the microsome assays (except when 3α-hydroxycostunolide was used as substrate; see above) we used a LC-LTQ-Orbitrap FTMS system

(Thermo Scientific) consisting of an Accela high performance liquid chromatograph (HPLC), an Accela photodiode array detector, connected to an LTQ/Orbitrap hybrid mass spectrometer equipped with an electrospray ionization (ESI) source. Chromatographic separation was achieved on an analytical column (Luna 3 μ C18/2 100 A; 2.0 × 150 mm; Phenomenex, USA). Eluent A (degassed) (HPLC grade water/formic acid (1000:1, v/v)) and eluent B (acetonitrile/formic acid (1000:1, v/v)) were used. The flow rate was set at 0.19 mL min$^{-1}$. The 45 min gradient was from 5 to 75% acetonitrile, followed by a 15 min washing step and equilibration. FTMS was done at a resolution of 60,000 while the resolution for MS$^n$ scans was set to 15,000. Calibration of FTMS was done externally in negative ionization mode by using sodium formate clusters in the range $m/z$ 150–1200, and automatic tuning was performed on $m/z$ 384.93. The injection volume was 5 μL.

For the analysis of product formation in the in planta assays, leaves were harvested 4.5 days post *Agrobacterium* infiltration (dpi) and snap-frozen in liquid N$_2$. Leaves were ground into a fine powder and extracted with 300 μl methanol/formic acid (1000:1, v-v) extraction buffer. Short vortexing was followed by 15 min of sonication. Then, extracts were centrifuged at 13,000 × *g* for 10 min and 200 μL of supernatant was passed through a 0.22 μm inorganic membrane filter (RC4, Sartorius, Germany) and were kept in injection vials for LC-Orbitrap-MS analysis as described above.

**Cysteine conjugation**. Cysteine conjugation was performed as described by Liu et al.[13]. In brief, cysteine (150 mM) in 7 μL potassium phosphate buffer (100 mM; pH 6.5) and standards (30 mM) in 7 μL ethanol were added to 1000 μL potassium phosphate buffer (100 mM; pH 6.5). The reaction was initiated by adding 7 μL of glutathione *S*-transferase (GST) (1 g L$^{-1}$, in 100 mM potassium phosphate buffer; pH 6.5) into the mixture. Complete assay mixtures with(out) GST enzyme or either of the substrates were used as controls which was analysed by LC-Orbitrap-FTMS as described above. Samples were incubated for 30 min at room temperature and were kept at −20 °C until analysis. Costunolide was purchased from TOCRIS Bioscience (United Kingdom). Parthenolide, 3β-hydroxycostunolide and 3β-hydroxyparthenolide, isolated from dried aerial parts of feverfew plants, were provided by Dr. Justin T. Fischedick from the PRISNA company[9]. Kauniolide, 1β,10β-epoxy-kauniolide (arglabin), 1α,10α-epoxy-kauniolide and 3β,4β-epoxy-kauniolide were kindly provided by Professor Yue Chen (State Key Laboratory of Medicinal Chemical Biology, Nankai University, China)[15].

**Chemical synthesis of 3α-hydroxycostunolide**. 1 mg of 3β-hydroxycostunolide was dissolved in 1 mL of pentane. Then, 50 mg of manganese dioxide (MnO$_2$, Sigma-Aldrich) was added and shaken at room temperature for 40 h. The supernatant was passed through a filter to remove MnO$_2$ and checked by GC-MS (see below). No 3β-hydroxycostunolide was detected in extracts and a new peak with the same mass as 3-oxocostunolide was detected. The solvent was evaporated to dryness by gently blowing a stream of N$_2$. In order to reduce 3-oxocostunolide, the sample was dissolved in 1 mL of ethanol and 10 mg of sodium borohydride (NaBH$_4$) was added. The mixture was shaken for 1 h at room temperature. Then, 1 drop of acetic acid was added to the mixture to destroy the complex of the product with borium and liberate the free alcohol. The solvent was then evaporated with a vacuum concentrator and the residue dissolved in 1 mL diethyl ether. Subsequently, 1 mL of saturated NaHCO$_3$ was added to the mixture to remove the acetic acid and the sample was washed several times with 1 mL of MQ water. The mixture was then extracted two more times with 1 mL diethyl ether, dried over Na$_2$SO$_4$, concentrated and injected into GC-MS.

**GC-MS analysis**. A gas chromatograph (7809A, Agilent, USA) equipped with a 30 m × 0.25 mm × 0.25 μm film thickness column (DB-5) was used. Helium was used as the carrier gas and the flow rate was adjusted to 1 mL min$^{-1}$ for GC-MS analysis. No splitting was used for injection and inlet temperature set to 250 °C. The initial oven temperature was 45 °C for 1 min, and increased to 300 °C after 1 min at a rate of 10 °C min$^{-1}$ which was held for 5 min at 300 °C. The GC was coupled to a Triple-Axis detector (5975C, Agilent).

**Homology modelling**. To create homology models of kauniolide synthase, costunolide 3β-hydroxylase and parthenolide synthase, tertiary structure templates were sought using the Protein-Protein Basic Local Alignment Tool (BLASTP) from the National Center for Biotechnology Information (NCBI), focussing on the PDB database[40]. Additional templates were retrieved using Protein-Protein HMMER (phmmer), directed to all available databases[41]. The resulting query-template alignments were compared in the Multiple Sequence Viewer of the software package Prime[42,43] within Maestro (https://www.schrodinger.com/maestro). One template hit was chosen based on sequence identity, residue homology and the presence of a ligand. Substrate Recognition Sites (SRS) (which were identified using the Cytochrome P450 Engineering Database (CYPED)[44]) were checked thoroughly for homology. A few cycles of homology model building and manual realignment followed. During the realignments, gaps/insertions were shifted in regions of low homology to connect residues smoothly with regions of higher homology. The N-terminal α-helix is absent in the model as none of the templates covered this region in their X-ray structure. Supplementary Table 2 lists the final query-template

alignments. The homology models were cleaned up: excess waters/ions from the template structure were removed, appropriate charges were applied to the residues and disulphide bonds of cysteines were created using the Protein Preparation Wizard of Prime[42,43]. The haem was inherited from the template but was replaced by a quantum mechanically described haem from the Parthenolide synthase model. For this quantum mechanical model of the haem, Compound I was created (Fe$^{2+}$ was replaced by Fe$^{5+}$ and O$^{2-}$ was bonded to Fe$^{5+}$) in Maestro. Costunolide was docked in this modified Parthenolide synthase before QSite[42,43] (https://www.schrodinger.com/qsite), and a mixed quantum mechanics/molecular mechanics software (QM/MM) was used on this system. The molecular mechanic region was comprised of the protein plus the substrate without the cysteine that bonds with the iron centre. The quantum mechanical region was comprised of Compound I and the iron-chelating cysteine. For the quantum mechanical description, the DFT-MO6 function set was chosen. The OPLS2005 in vacuo Force Field was set for the molecular mechanical calculations. QSite[45,46] kept the substrate and residues beside the cysteine frozen during the calculations. After inserting the haem from Parthenolide synthase in kauniolide synthase and costunolide 3β-Hydroxylase, the homology model was minimized. Then, 500 cycles of OPLS2005 Force Field minimization were run using Impact under Maestro (https://www.schrodinger.com/maestro). Implicit water was included in the minimization acting as an Analytic Generalized Born solvent. The resulting structure was used in substrate–enzyme interaction modelling.

**Substrate modelling**. Costunolide was drawn in Chem3D (Version 14.0.0.117) and minimized with the OPLS2005 Force Field over 100 cycles in Impact from Maestro. Implicit water was modelled during the minimization by the Poisson Boltzmann Solver. After this molecular mechanics minimization, costunolide was described quantum mechanically with Jaguar 44 (https://www.schrodinger.com/jaguar) using the B3LYP-DFT level of theory and the 6-31G** basis set. Electrostatic potential surface atomic charges were then extracted and used for the following enzyme–substrate interactions.

**Enzyme–substrate interaction modelling**. In preparation of binding simulations, costunolide was docked in KLS, costunolide 3β-hydroxylase and PTS. Hereto, Glide[47,48] was used to produce a receptor grid of the active site, using as centre the position of the ligand in the homology template. Polar side chains were allowed to rotate in the grid. Glide was used to dock costunolide in extra precision mode, using the partial charges from the quantum mechanical description of costunolide. A total of 5000 Initial docking poses were produced. The 100 best poses were then minimized. One final docking pose was returned by Glide. This system was used as input for a molecular dynamics simulation using Desmond 49-50. Explicit water molecules with 150 mM of sodium chloride were simulated in a box of 10 Å surrounding the protein. A NPT system was maintained, providing a constant number of particles, pressure and temperature. Standard pressure (~1 bar) and a temperature of 300 K were applied. A 100 N m$^{-1}$ constraint was set to Compound I and a 50 N m$^{-1}$ constraint was set onto the protein backbone. The simulation duration was 5 ns maximally. System volume and energy were inspected afterwards using the Quality Analysis tool of Desmond[49,50]. At a potential energy minimum as close to the end of the simulation as possible, one simulation frame was selected. Compound I from the first simulation frame was inserted in the selected final simulation frame. For kauniolide synthase and parthenolide synthase this simulation frame was used as input for a subsequent simulation with PELE[22,23], aiming at refining the binding mode of costunolide in the CYP450s. The costunolide 3β-hydroxylase with docked costunolide was used as input directly. A force constant of 400 N m$^{-1}$ was set on the bond of 2.45 Å between the iron-chelating cysteine sulphur and iron cation.

PELE generates a MC chain where each step uses a ligand and backbone perturbation followed by a relaxation step involving a side chain prediction and a full-system minimization. Each ligand perturbation involved 50 different independent trials with: (i) a rotational change of 0.02 radian (70% probability) or 0.25 radian (30% probability); (ii) a translational displacement of 1.0 Å (70% probability) or 0.5 Å (30% probability). The backbone perturbation followed an anisotropic network model approach using the lowest 6 eigenvectors. All side chains within 6 Å from the substrate were included in the relaxation step. PELE ran on the system for maximally 5000 steps. Two different set of simulations were run: one without a constraint on the substrate and one with a restricted search region where the substrate was not allowed to move beyond 15 Å from Compound I. To characterize the enzyme–substrate interactions, PELE used the 2005 OPLS-AA force field and a surface-SGB implicit model solvent.

## Data availability
The cDNA sequence of the candidate genes has been submitted in GeneBank under the primary accession numbers MF197558 (Tp8879 (kauniolide synthase)) and MF197559 (Tp8886 (kauniolide oxidase)). The sequences were also deposited in David Nelson's cytochrome P450 database (http://drnelson.uthsc.edu/cytochromeP450.html); Tp8879 and Tp8886 were assigned the names CYP71BZ6X and CYP71DD5, respectively[36]. The RNA-seq dataset (www.terpmed.eu) and computer codes used for this study are available from the corresponding author on reasonable request.

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

## Acknowledgements

We would like to thank Dr. Justin T. Fischedick and Professor Yue Chen for providing sesquiterpene lactone standards. Q.L. was funded as a part of Terpmed (Plant Terpenoids for Human Health: a chemical and genomic approach to identify and produce bioactive compounds) (Project ID 227448) and A.B.K. was funded by Iranian Ministry of Science Research and Technology.

## Author contributions

A.B.K., Q.L., A.v.d.K. and H.B. designed the research experiments. Q.L., A.B.K., L.R., D. M. and I.P. performed gene cloning. D.M. did the gene expression analyses by qPCR. Q. L. and L.R. did transient expression in *N. benthamiana*, yeast microsome assays and LC-Oribitrap-FTMS. A.B.K. performed yeast transformation, chemical synthesis, GC-MS analysis, analysed assembled RNA-seq expression data, phylogenetic and transcript analysis. A.B.K., A.v.d.K., V.G. and M.F.L. designed the modelling experiments and P.K. conducted protein docking. Q.L., A.B.K., D.M., I.P., L.R., P.K., M.F.L., R.d.V., M.C.R.F., A.v.d.K and H.B. interpreted the data and reviewed the manuscript. A.B.K. and Q.L. wrote the manuscript.

## Additional information

**Competing interests:** The authors declare no competing interests.

