## [Peer Review File · Nature Communications]

Reviewers' comments:

Reviewer #1 (Remarks to the Author):

The authors present a thorough investigation into a novel biosynthetic pathway of a particular compound, kauniolide, by P450 enzymes in feverfew. The compound itself is of some pharmaceutical interest. The elucidated novel enzymatic activities may furthermore hold interest for biosynthesis routes for other products. Finally, the methodology applied is not novel, but different techniques are used in a way that complements each others' strengths, which may be a good starting point for similar future studies.

The results presented, mainly MS supplemented by strategically applied molecular modelling, provide a strong basis for the conclusions about the biosynthetic routes. In this, it constitutes a solid step forward in our knowledge of this rather important area of biosynthesis of pharmaceutically active compounds. Methods are explained in sufficient detail to allow reproduction by a capable researcher.

The section about the 'Functional characterization of selected Cyp71s in yeast' (1132-1165) deals mainly with Tp8879, and provides quite some experimental detail about its activity. At the end, suddenly 'Moreover, Tp8886' is mentioned as acting on several compounds, but no experimental data or literature references are provided. This seems strange, and should be easily remedied.

Below I list some minor concerns, which I believe if addressed may improve the accessibility of the paper:

- the abstract is quite thick in organic chemistry jargon, this may be reduced to increase the attractiveness for non-experts in that particular area.
- 158: '(reviewed in 9)', I believe it is bad writing style to use a reference number as noun.
- 166: there may be a comma missing before 'chicory'
- 198: the abbreviation 'KLS' is not introduced (TpKLS is, but this I believe can be confusing)
- 1113: the comma should be after 'though', not before.
- 1126: 'For the those', here something seems to be wrong in the sentence.
- 1138: the notation '[M+H]' should be introduced for the non-MS-initiates.
- 1148: there seems to be a superfluous set of sharp '[''] brackets, inbetween round '()' brackets at the end of this sentence.
- 1191-1192: 'hydroxylation of costunolide at the C3 of (Fig. 3e).' I do not understand this reference to the MS spectrogram shown there. There is no C3 indicated. Please clarify. (Perhaps 5e is meant?)
- 1200-1203 'Supposedly, in those cases the main product of TpKLS.' It seems something is wrong in the construction of this sentence, or I may be missing something, so I cannot understand it well. Please clarify.
- 1238+1240: two times 'However' is not very good form, and as this is close to the main point of the paper I suppose the authors may wish to remedy this.
- 1272-1273: similarly, 'However, ... nevertheless ...' doesn't read well either.
- 1322: 'ESTs' is not introduced.
- p26 (1618-1630): the literature references appear without superscript several times.

Figure 1:

- the arrows from germacrene A acid to costunolide, and from costunolide to kauniolide cross through the boxes, respectively another compound. This makes the figure harder to read. It would be good if this could be cleared up.
- The meaning of the grey background several of the compounds is not explained.

Figure 5:

- the three-D diagrams in b and c look very pretty, but are actually hard to read. The colors are helpful, but the perspective view does not help, and the transparency of the cylinders makes it

look messy. Perhaps a simpler 2D plot, like a heatmap, might be easier to digest.

- In the panels d and e, the projection is orthographic (not perspective), which leads to loss of an indication of depth. In addition, or alternatively, some shadowing or other depth-cueing may be used. This will make the figure much easier to take in a single glance.

Reviewer #2 (Remarks to the Author):

The authors here report the biosynthesis of kauniolide from constunolide, and for the first time show that an enzyme can convert constunolide into a guaianolide. The paper clearly present the data that support this result. From both yeast expression and in planta data.

Unfortunately the authors miss to show what is the mechanism of this enzyme. The authors show that they can produce 3-beta-hydroxy-costunolide enzymatically. With this tool in hand one could produce sufficient amount of the 3-beta that through first a reaction with activated MnO₂, will get the allylic alcohol that will oxidize to the ketone. With a mild reducing agent (NaBH₄) one should get both the alpha and beta alcohols. These can be separated and one then have the 3-alpha alcohol to test in the enzyme assay. Don't use LiAlH₄, since this will reduce both the ketone and the lactone. The authors should perform this test prior to any publication of any hypothetical reaction mechanism. This will confirm whether figure 7 is anywhere near the truth.

I'm aware that the in silico trials suggest this, but the P450 model is based on a human+rabbit crystal, and from many other papers published on P450's it is well established that plant P450's do not really fit this crystal. Thus, the modelling is not conclusive like the authors would like it to be.

Testing of the enzyme with 3-alpha will could maybe also establish whether the reaction mechanism proceeds through the suggested protonation (as suggested), or via a bi-hydroxy form that spontaneously rearrange to a ketone. With a slight acidic group close to the ketone this will act as the "carbocation" establishing group and allow for the ring closure. But this is all speculations since the 3-alpha group was not tested. But if 3-alpha is not a substrate, well what is then the mechanism. This remains to be elucidated. The current suggestion is wild speculations and should be supported by biochemical data.

Along with this the authors wish to report the characterizations of Tp8886 - again the authors show that this is expressed in yeast, thus it must be a matter of scale up to get enough material to perform a NMR on the product. The current status of this enzyme should not be published prior to this. Costunolide can be purchased rather cheap, thus the cascade reaction of TP 8879 and Tp8886 can also be performed to show that these are connected.

These are the two major remarks and should be fixed prior to any publication.

Other remarks. Please do provide all CYP numbers based on the official CYP nomenclature. Through the manuscript this is mentioned for some but not all, and this is a rather easy task to fix prior to publication. But this will significantly ease any future studies in this area - especially with the phylogenetic studies of P450's. Please provide this for alle feverfew CYP's and others mentioned in the text.

It was recently published that (in Apiaceae) that the terpene synthases and the p450's are co-localized in the plant. It this also true in feverfew, and has in-situ pcr been performed in feverfew? This would strengthen the co-expression studies.

There are references to Figure 2c and 2d, but these are not given in the uploaded text.

line 149 - what is the rt 10.7? or is it 10.07?

line 150 according to??? sentence missing.

Maybe Tp8879 is not so unique after all, and establishment of the carbocation or ketone would also in slightly acidic conditions lead to a possible ring closure.

Reviewer #3 (Remarks to the Author):

The authors report within the isolation of a Cyp71 cytochrome P450 from *Tanacetum parthenium*, kauniolide synthase (TpKLS or Tp8879) which performs an elusive biosynthetic transformation to afford a guaianolide from a germacranolide, specifically kauniolide from costunolide, a transformation requiring hydroxylation and cyclization reactions.

This work provides strong support for a single enzyme (Tp8879) alpha-hydroxylating C3 of costunolide and a subsequent cyclization event to afford kauniolide. This enzymatic activity of Tp8879 towards costunolide was confirmed by in vivo testing and subsequent product analysis. LC-Orbitrap-FTMS showed formation of a new product possessing a mass reduction of 2.015 D when compared to costunolide. Conjugation of this unknown compound with cysteine produced a new compound that matched a co-injected sample of kauniolide-cysteine eluting at the same retention time and having the same mass. The biosynthetic pathway was reconstructed in *Nicotiana benthamiana*, a model organism, and a compound with the same exact mass and retention time as kauniolide-Cys adduct was found in the leaf extracts. Enzymatic action of TpKLS is stereoselective in that only 3 alpha-hydroxycostunolide is converted to kauniolide and not the 3 beta-hydroxycostunolide isomer. In silico docking experiments were performed showing a favorable distance between the heme oxy-anion in the TpKLS model and the alpha C3sp³-H. In all cases control experiments were performed using parthenolide and oxidation products were not produced.

This work is significant because it has been assumed that guaianolides are derived from germacranolides but a biosynthetic pathway has not been forthcoming. Furthermore, elucidation of KLS and a reconstructed biosynthesis of kauniolide in *Nicotiana benthamiana* has afforded the first committed intermediate in guaianolide synthesis.

The PI is an expert in sesquiterpene lactones and phenolic diterpenes and their signaling activity in plants.

The results reported within will be valuable to those working in the general area of sesquiterpene lactones and those practicing small molecule synthesis either through traditional methods or through metabolic engineering.

Minor changes are recommended:

Figure 1-my version is missing the diphosphate group from farnesyl diphosphate, Micheliolide is misspelled, all stereochemistry should be depicted on guaianolide structures. Figure 1 needs to be discussed a bit more in the text of manuscript.

Figure 2- the heat map legend does not have units

Please find below our detailed responses to the reviewers' comments and suggestions. All changes have been marked in the revised manuscript.

Reviewer #1:

- Comment: The section about the 'Functional characterization of selected Cyp71s in yeast' (1132-165) deals mainly with Tp8879, and provides quite some experimental detail about its activity. At the end, suddenly 'Moreover, Tp8886' is mentioned as acting on several compounds, but no experimental data or literature references are provided. This seems strange, and should be easily remedied.
- Response: For clarity and to improve the focus of the manuscript, we have excluded Tp8886 from the manuscript.

- Comment: The abstract is quite thick in organic chemistry jargon, this may be reduced to increase the attractiveness for non-experts in that particular area.
- Response: - We have reduced the 'organic chemistry jargon' as much as possible.

- Comment Figure 1: the arrows from germacrene A acid to costunolide, and from costunolide to kauniolide cross through the boxes, respectively another compound. This makes the figure harder to read. It would be good if this could be cleared up. The meaning of the grey background several of the compounds is not explained.
- Response: Figure 1 was adapted according to the reviewer's suggestions.

- Comment Figure 5: The three -D diagrams in b and c look very pretty, but are actually hard to read. The colors are helpful, but the perspective view does not help, and the transparency of the cylinders makes it look messy. Perhaps a simpler 2D plot, like a heatmap, might be easier to digest.
- Response: The Figure was changed to a 2D plot, showing the binding energy and occasions in which substrate was docked into the active site. Colours are provided as heatmap.

- Comment Figure 5: In the panels d and e, the projection is orthographic (not perspective), which leads to loss of an indication of depth. In addition, or alternatively, some shadowing or other depth-cueing may be used. This will make the figure much easier to take in a single glance.
- Response: a dark background was added to the Figure to obtain a better perspective.

- Comment - 158: '(reviewed in 9)', I believe it is bad writing style to use a reference number as noun.
- Response: the sentence was adapted.

- Comment - 166: there may be a comma missing before 'chicory'
- Response : Adapted.

- Comment - 198: the abbreviation 'KLS' is not introduced (TpKLS is, but this I believe can be confusing).
- Response: KLS upon first use is introduced in the introduction : ‘Unexpectedly, in these assays we identified a Kauniolide Synthase (KLS), a single P450 enzyme able not only to form hydroxylation activity but also water elimination coupled to cyclisation and regioselective deprotonation, leading to the synthesis of kauniolide ’

- Comment - 1113: the comma should be after 'though', not before.
- Response: Adapted.

- Comment - 1126: 'For the those', here something seems to be wrong in the sentence.
- Response: Adapted.

- Comment - 1148: there seems to be a superfluous set of sharp '['] brackets, in between round '()' brackets at the end of this sentence.
- Response: Corrected.

- Comment- 1191-192: 'hydroxylation of costunolide at the C3 of (Fig. 3e).' I do not understand this reference to the MS spectrogram shown there. There is no C3 indicated. Please clarify. (Perhaps 5e is meant?).
- Response: Sentence was changed to ‘The production of 3 β -hydroxycostunolide, though as a minor product, from feeding costunolide to *TpKLS* suggests that this enzyme can perform hydroxylation of costunolide at the C3 position.’

- Comment:- 1200-203 'Supposedly, in those cases the main product of TpKLS.' It seems something is wrong in the construction of this sentence, or I may be missing something, so I cannot understand it well. Please clarify.
- Response: Sentence was changed to 'The occasional release of the intermediate 3 β -hydroxycostunolide from the enzymatic cavity would explain the presence of 3 β -hydroxycostunolide as side product of *TpKLS*.'

- Comment- 1238+240: two times 'However' is not very good form, and as this is close to the main point of the paper I suppose the authors may wish to remedy this.
- Response: Done.

- Comment - 1272-273: similarly, 'However, ... nevertheless ...' doesn't read well either.
- Response: Adapted.

- Comment- 1322: 'ESTs' is not introduced.
- Response: Adapted.

- Comment- p26 (1618-630): the literature references appear without superscript several times.
- Response: Corrected.

Reviewer #2

According to the suggestions of reviewer #2 we have chemically synthesized 3 α -hydroxycostunolide and performed the enzymatic reactions. The results of this experiment are now provided in Figure 3. These results support the proposed mechanism of action of TpKLS, which is now provided in Figure 7. Reviewer #2 also raised some doubts about the modelling experiments and the homology models. We would like to mention that sequence identity and coverage were highest on the selected relevant areas (beyond the known variable regions of P450s), with the selected templates from human and rabbit origins. The sequence identity within these selected regions were even higher than with other plant P450s. The low homology with other plant P450s could mean that these are quite specialized P450s, which especially for KLS seems a reasonable assumption, considering its remarkable activity. To emphasize the limitations of the modeling studies we added the sentence: 'We note that

modelling results are based on a heterologous P450 template model. Future acquisition of the *TpKLS* crystal structures may substantially improve the in-silico simulation.'

- Comment: Reviewer #1 and #2 asked about the product identification of *Tp8886* enzyme.
- Response: due to very low abundance of these molecules, we could not identify these products. For clarity and focus of the manuscript, *Tp8886* has been omitted.

- Comment: Reviewer #2 also asked about the co-localization of feverfew terpene synthases and cytochrome P450s. From the comment it was not clear whether this question was about colocalisation of genes on the genome or proteins within the same cells.
- Response: the cDNAs of the TPS and P450 were from the same trichome enriched cDNA library. No in-situ PCR on transcript of these genes in feverfew has been done. The following sentence was added to the methods section: 'Both the TPS and P450 cDNA sequences used in this study were identified in the same glandular trichome enriched cDNA library and we assume that both genes are indeed expressed in the same cell'.

- Comment: There are references to Figure 2c and 2d, but these are not given in the uploaded text.
- Response: Text was corrected.

- Comment: what is the rt 10.7? or is it 10.07?
- Response: RT is indeed 10.07. Corrected.

- Comment: according to??? sentence missing.
- Response: Sentence corrected : 'The product eluting at RT=10.07 was identified as 3 β -hydroxycostunolide-cysteine (Fig. 2d and 2e) according to Qing et al., (2014)¹³.'

Reviewer #3:

- Comment Figure 1-my version is missing the diphosphate group from farnesyl diphosphate, Micheliolide is misspelled, all stereochemistry should be depicted on guaianolide structures.
- Response: the diphosphate group has been added to the farnesyl diphosphate structure, micheliolide spelling was corrected and stereochemistry is now shown.

- Comment Figure 2- the heat map legend does not have units
- Response: Heat map units were added to the figure and the figure was transferred to the supplementary information.

Reviewers' comments:

Reviewer #1 (Remarks to the Author):

I am satisfied with the response to my previous issues, and the amendments to the manuscript and figures.

As a note, the black background in Fig 3 seems to shine through the white hydrogens.

Reviewer #2 (Remarks to the Author):

The manuscript and work is very much improved. Thus the comments below is more seen as a major revision, since part of the experimental is missing. Overall the method description needs an overhaul to clarify what parts has been used where. It is not clear how the different parts have contributed to the presented results and the reader is left with some guess work. References to e.g. figures that present the data from the individual experiments would be crucial. Also linking the methods - e.g. the extract of this assay was examined by LC-MS.

Also I still have a major problem with the suggested protonation. Regularly we see that P450s perform multiple hydroxylations. Could this be the case here, that there is a triple hydroxylation that instead of acid leads to dehydroxylation and establish a carbocation that can then act towards the ringclosure?

Or maybe an epoxy with c-4/2?

The proposed mechanism is not seen before for P450's and I recommend the authors to remove this suggestion from 7. There is nothing in the data that suggest this pathway and it is pure speculations.

I will also recommend the authors to perform further studies (in an other publication), like specific changes of aminoacids. This could perhaps give hints towards the mechanism. Labelling could also be performed, though this is tricky with carbocation formation etc.

Other recommendations

Line 52: It is not true that ALL sesquiterpene lactones contain a α -methylene- γ -butyrolactone, most do yes, but Artemisinin does not, just as an example of one of the many that have a α -Methylene- γ -pentalenolactone. Please adjust accordingly. It is true that the α - γ lactone accounts for the activity, but as clarified in many reviews this is not the only source. It is more likely that the general activity is from the α , β -unsaturated carbonyl. As described for guainolides (e.g. in here https://link.springer.com/referenceworkentry/10.1007%2F978-3-642-22144-6_134)

Line 97: /cuclysatation - should be cyclisation I guess.

Line 222: What is EV??? this is not explained anywhere in the manuscript. Likewise a major problem is that the enzyme assay described for the GC study is not described in the methods. Thus the reader do not stand a chance to reproduce this crucial experiment. Either this is hidden somewhere in the methods, but I suspect this has not been included, by mistake of cause.

Figure 1: Remove the arrow between the beta-3-hydroxycost to kaunolide. Clean up the figure so this is IN-line with your results, the current figure mislead the authors even though the rest of the text is clear, but figures are shared much more easy than the text. Thus try not to present things you show are not happening. Also do not include the speculative protonation.

You can always discuss the protonation in the text, but I still do not see the evidence for this suggestion.

So apart from that, it is really a good story and I hope the authors will make more lean. ;) And it is now clear that 3- α is the middle substrate towards the kaunolide.

Reviewer #3 (Remarks to the Author):

I am completely satisfied with the changes that have been made by the author and recommend publication in Nature Communications.

Reviewer#1:

- Comment: As a note, the black background in Fig 3 seems to shine through the white hydrogens.
- Response: The figure quality was improved.

Reviewer #2

- Comment: Overall the method description needs an overhaul to clarify what parts has been used where. It is not clear how the different parts had contributed to the presented results and the reader is left with some guess work. References to e.g. figures that present the data from the individual experiments would be crucial. Also linking the methods - e.g. the extract of this assay was examined by LC-MS.

Response: The organisation of the methods section has been improved.

- Comment: Also I still have a major problem with the suggest protonation. Regularly we see that P450s perform multiple hydroxylations. Could this be the case here, that there is a triple hydroxylation that instead of acid leads to dehydroxylation and establish a carbocation that can then act towards the ringclosure? Or maybe an epoxy with c-4/2? The proposed mechanism is not seen before for P450's and I recommend the authors to remove this suggestion from 7.

Response: Indeed P450s are known to be hydroxylating/epoxidising enzymes and there is no doubt that kauniolide synthase starts its action by the alpha-hydroxylation of costunolide. However, after that, this extra oxygen atom is lost and the ONLY (bio)chemically feasible way is by protonation, leading to water (as a good leaving group) and a resonance-stabilised carbocation. Another hydroxylation, as the reviewer suggests, would lead to even more oxygen atoms inside the product, which is in contrast to our findings. Epoxidation at the C4-C5 double bond does give guaianes after ring closure, but the products still contain an oxygen atom because the epoxide is turned into a hydroxyl group (see Piet et al., Tetrahedron 1995, 51, 6303). In conclusion, the protonation step is crucial and unavoidable in the explanation of the formation of kauniolide. For this reason we wish to keep that step in the text and figures. Indeed, this is a novel action of a P450, as a follow-up of the usual oxygenation reaction, but that is exactly the reason why we think this paper is interesting for a broad audience and will stimulate others to look for similar activities in other P450s.

- Comment: Line 52: It is not true that ALL sesquiterpene lactones contain a α -methylene- γ -butyrolactone, most do yes, but Artemisinin does not, just as an example of one of the many that have a α -Methylene- γ -pentalenolactone. Please adjust accordingly. It is true that the alfa-gamma lactone accounts for the activity, but as

clarified in many reviews this is not the only source. It is more likely that the general activity is from the α , β -unsaturated carbonyl. As described for guainolides (e.g. in here https://link.springer.com/referenceworkentry/10.1007%2F978-3-642-22144-6_134).

Response: The sentence has been adapted and the above mentioned reference added 'Natural occurring sesquiterpene lactones often contain an α -methylene- γ -butyrolactone and/or α,β -unsaturated cyclopentenone moiety which together account for their biological activity against cancer and inflammation^{3,4}.

- Comment: Line 97: /cuclysatation - should be cyclisation I guess.

Response: Corrected.

- Comment: Line 222: What is EV??? this is not explained anywhere in the manuscript.

Response: EV means empty vector (control) and this is now explained throughout the manuscript.

- Comment: Likewise a major problem is that the enzyme assay described for the GC study is not described in the methods. Thus the reader do not stand a chance to reproduce this crucial experiment. Either this is hidden somewhere in the methods, but I suspect this has not been included, by mistake of cause.

Response: Corrected and added to the methods section as 'Yeast microsome in vitro assays for feeding 3α -hydroxycostunolide was performed as described above, but after 2.5 hr incubation at 25°C and shaking at 200 rpm, samples were extracted 3 times with 2 ml ethyl acetate. The ethyl acetate phase was passed through a pasture pipet, plugged with glass wool and filled with Na₂SO₄ to dehydrate the extract. The extract was then concentrated and injected into GC-MS.'

- Comment: Figure 1: Remove the arrow between the beta-3-hydroxycost to kauniolide. Clean up the figure so this is IN-line with your results, the current figure mislead the authors even though the rest of the text is clear, but figures are shared much more easy than the text. Thus try not to present things you show are not happening. Also do not include the speculative protonation.

Response: Figure 1 has been cleaned up. We propose to keep the proposed protonation as discussed above.

We hope that with these changes our manuscript is now acceptable for *Nature Communications*.

Prof. Dr. H.J. Bouwmeester

Swammerdam Institute for Life Sciences

REVIEWERS' COMMENTS:

Reviewer #2 (Remarks to the Author):

The manuscript is truly much better now. I accept the chemical suggestion, but I still would not be comfortable with the figure suggestion. Merely because figures are shared super fast these days and easy becomes the "truth". I will recommend the manuscript is published now. Great work by the way.

Response to Referees:

Reviewer 2 has one additional issue: Figure 7. Instead of removing this Figure, we have adapted the legend such that the readers can clearly see that the proposed protonation is hypothetical and has not been proven in the present study. With this adaptation, we can still use our proposed mechanism of action in the manuscript. These and all other changes we needed to make according to the checklists have been marked in the revised manuscript.

With kind regards,

Prof. dr. H.J. Bouwmeester
Swammerdam Institute for Life Sciences